# Walking Trajectory Estimation Using Multi-Sensor Fusion and a Probabilistic Step Model

**DOI:** 10.3390/s23146494

**Published:** 2023-07-18

**Authors:** Ethan Rabb, John Josiah Steckenrider

**Affiliations:** 1School of Mechanical Engineering, Purdue University, West Lafayette, IN 47907, USA; 2Department of Civil and Mechanical Engineering, United States Military Academy, West Point, NY 10996, USA; john.steckenrider@westpoint.edu

**Keywords:** outdoor localization, sensor fusion, map estimation, human gait measurement

## Abstract

This paper presents a framework for accurately and efficiently estimating a walking human’s trajectory using a computationally inexpensive non-Gaussian recursive Bayesian estimator. The proposed framework fuses global and inertial measurements with predictions from a kinematically driven step model to provide robustness in localization. A maximum a posteriori-type filter is trained on typical human kinematic parameters and updated based on live measurements. Local step size estimates are generated from inertial measurement units using the zero-velocity update (ZUPT) algorithm, while global measurements come from a wearable GPS. After each fusion event, a gradient ascent optimizer efficiently locates the highest likelihood of the individual’s location which then triggers the next estimator iteration.The proposed estimator was compared to a state-of-the-art particle filter in several Monte Carlo simulation scenarios, and the original framework was found to be comparable in accuracy and more efficient at higher resolutions. It is anticipated that the methods proposed in this work could be more useful in general real-time estimation (beyond just personal navigation) than the traditional particle filter, especially if the state is many-dimensional. Applications of this research include but are not limited to: in natura biomechanics measurement, human safety in manual fieldwork environments, and human/robot teaming.

## 1. Introduction

### 1.1. Background

Human localization is an active research topic with many applications, from the accurate determination of human position in teams of humans and mobile robots to the study of human biomechanics. The variety of fields implementing human–robot teams includes agriculture [1], mining [2], search-and-rescue [3], and exploration [4]. Each sector has its own requirements for safe and effective human–robot teams due to the different environments and levels of human involvement. Accurate personal navigation systems for human motion make for safer human–robot teams because they allow robots to determine where their human counterparts are so that they can avoid and work around them. While there are currently no specific Occupational Safety and Health Administration (OSHA) standards for robotics in the workplace, there have been efforts to identify risks and hazards and apply them to human–robot teams [5]. ISO 15066 identifies standards for robotic arms in human–robot teams [6]; however, it does not establish such standards for mobile robots. In biomechanical applications, personal navigation systems assist in determining how a subject is moving. Parameters that may be determined from the accurate measurement of human motion include fatigue, injury, and other critical pathologies. To develop high-quality personal navigation systems, accurate estimation and localization methods must be developed.

In some applications, Gaussian estimation (i.e., the Kalman filter) is suitable and has sufficiently low computational requirements. However, in most nonlinear applications, such as the framework proposed here, the efficiency offered by Gaussian assumptions has an associated drawback in reduced estimation accuracy. Non-Gaussian estimation algorithms (e.g., particle and grid-based filters) tend to be more computationally expensive, which is a significant detriment in real-time navigational systems. However, these more complex estimation methods retain the information-rich probability distributions required for high-accuracy estimation. Although the techniques discussed here are applied to personal navigation, it is expected that the proposed non-Gaussian estimation framework can be generalized for application to a broad variety of contexts.

### 1.2. Related Work

There has been extensive work previously conducted on human localization via external sensors in controlled environments. Some approaches to estimate a walking human’s position in controlled environments include Ultra-Wide Band (UWB) [7], GPS [8], WiFi [9], and LiDAR sensors [10]. Each sensor has its own limitations, especially in uncontrolled environments, which can prevent accurate, non-intrusive, self-contained outdoor human localization.

In field environments where a human may cover great distances in potentially GPS- or satellite-denied areas, external sensors can become unreliable. As a result, wearable sensors are the preferred method of data collection because they are consistent and self-contained. Research has been conducted for personal navigation with IMUs mounted in various locations [11]. There have also been efforts to incorporate LiDAR into a wearable sensor for localization [12]. This LiDAR technique differs from the previously mentioned LiDAR in that it involves LiDAR mounted to an individual rather than LiDAR in a fixed position observing the individual. Due to the variety of sensors available, fusing sensor data are common in personal navigation.

Sensor fusion improves the accuracy and robustness of human localization by allowing the unique strengths of heterogeneous sensors to be combined such that estimation is improved. Sensor fusion often involves implementing filters to combine data from the sensors used. Particle filters (PFs) have been proposed to fuse IMU and UWB data for localization [13], while Kalman filters (KFs) have also been used to integrate IMU and GPS data [14]. Other sensor fusion combinations for human localization include fusing IMU data with visual odometry [15], cellular long-term evolution (LTE) data [16], and UWB data [17]. Outside of human localization, other techniques combine traditional GPS/IMU fusion with other sensors and models for ground vehicle localization [18,19,20]. Other Bayesian approaches combine algorithms with GPS/IMU fusion for UAV localization [21]. Neural network-based techniques are a non-Bayesian sensor fusion technique that have been applied to GPS/IMU fusion [22]. Beyond low-level sensor fusion, information fusion techniques range from deep learning methods [23] to even higher-level hybrid measurement and feature fusion [24,25]. Modern autonomous systems will continue to rely more and more heavily on such robust fusion technologies.

Oftentimes, sensors alone are not sufficient to provide accurate personal navigation because they can experience dropout, damage, aliasing, or any number of other issues. Recursive Bayesian estimation (RBE) is a framework which leverages a motion model of the object of estimation to enhance raw or fused sensor measurements and increase resilience to sensor unreliabilities [26]. RBE is a generalized technique which makes no assumptions about the structure of the probability distribution functions (PDFs) underlying state belief. The stages of RBE include prediction, observation (or measurement), and correction (or updating) [27]. RBE has many variants, three of which are discussed below.

The aforementioned Kalman filters, particle filters, and grid-based filters (GBFs) are examples of specific implementations of RBE which make varying assumptions and approximations. KFs assume state belief is Gaussian and can therefore be propagated via mean vectors and covariance matrices [28]. PFs are computationally more expensive than KFs, but they retain more information about the state when it is highly non-Gaussian. They function by propagating many points (particles) through a motion model, weighting them according to a sensor observation, and retaining the points with the highest probability of representing the actual state [29]. PFs are well-suited for low-dimensional nonlinear and non-Gaussian problems, but they become inefficient in many dimensions. Grid-based filters are less common estimators which sample state PDFs at regular grid cells and propagate the PDFs by performing discrete RBE operations on each cell [30]. While the PF and GBF do not make the Gaussian assumption of the KF and can therefore offer higher accuracy in complex estimation scenarios, they have additional computational requirements. Another non-Gaussian estimation method is maximum *a posteriori* (MAP) estimation, which involves finding the mode or modes of a state belief distribution via gradient methods, random sampling, or expectation–maximization [31]. Previous research in human localization has incorporated non-Gaussian estimators, but such research has been limited by controlled indoor environments [32].

The current methods which exist for human localization typically either rely on an extensive and obtrusive sensor suite or make various assumptions in estimation which sacrifice accuracy or efficiency. There is a need to develop a real-time solution to accurately estimating a human’s location during extended outdoor walking without making too many assumptions or sacrifices.

### 1.3. Objectives and Outline

The work presented here offers a solution to personal navigation which incorporates wearable and external sensors, information fusion, and a non-Gaussian motion model to improve human trajectory estimation in outdoor environments. The original contributions of this research include: (1) a statistically-driven gait model for incorporation into an RBE framework, and (2) a new maximum a posteriori (MAP)-type technique for non-Gaussian estimation which offers greater efficiency than traditional approaches. Although MAP estimation is generally a thoroughly researched field, this work presents a novel application of MAP to personal navigation via GPS and IMU fusion.

This paper is organized as follows. Section 2 establishes some fundamental principles underlying the original contributions and affiliated aspects of the research, while Section 3 details these contributions. Section 4 presents results from a Monte Carlo simulation study, and finally Section 5 offers some conclusions and future work.

## 2. Methods

This section describes the fundamental concepts required for understanding non-Gaussian estimation and GPS/IMU fusion for human localization as applied in this article.

### 2.1. Recursive Bayesian Estimation

Recursive Bayesian estimation (RBE) is a generalized probabilistic framework which underlies most common estimators and filters. RBE consists of prediction, observation, and correction stages, each of which probabilistically transforms information about a system’s state, $x_{k}$, from one time step *k* to the next. This state information, or belief, is represented mathematically by probability distribution functions (PDFs) notated by $p(x_{k})$. The three stages of RBE are described and formulated in the following sections.

#### 2.1.1. Prediction

The prediction stage of RBE leverages a system’s motion model to propagate state belief from one time step to the next. This stage is governed by the Chapman–Kolmogorov equation, given as follows:(1)$$\begin{array}{c} p\left(x_{k}|z_{1:k-1}\right)=\int_{X}p\left(x_{k-1}|z_{1:k-1}\right)p\left(x_{k}|x_{k-1}\right)dx_{k-1}. \end{array}$$

In Equation (Equation 1), $p(x_{k-1}|z_{1:k-1})$ is the PDF corresponding to state belief at step k−1 given all prior observations $z_{1:k-1}$. The state transition PDF, $p(x_{k}|x_{k-1})$, dictates the belief about the state at step *k* given the former state. This distribution is closely linked to the system’s motion model and governs how the state evolves. For a Markovian process with independent and identically distributed increments, this PDF becomes just $p(x_{k}-x_{k-1})$. Consequently, the Chapman–Kolmogorov equation becomes a convolution integral for such processes. The output of prediction is $p(x_{k}|z_{1:k-1})$, henceforth referred to as the predicted PDF.

#### 2.1.2. Observation

Observation is usually accomplished by some real-time sensing or measurement modality. This stage of RBE is responsible for mathematically transforming a single measurement at step *k*, $z_{k}$, into an observation likelihood $l(x_{k}|z_{k})$. This likelihood function describes the distribution of belief about the true state given the received observation. The specifics of this stage are typically obtained by probabilistic sensor or observer characterization and are therefore context-dependent.

#### 2.1.3. Correction

The third stage of RBE combines information from the prediction and observation stages to update belief about the state being estimated. Correction, also known as Bayesian inference, belief fusion, or simply updating, is accomplished by multiplying the predicted PDF with the observation likelihood and normalizing the resulting distribution. Mathematically, this is expressed as
(2)$$\begin{array}{c} p\left(x_{k}|z_{1:k}\right)=\frac{l\left(x_{k}|z_{k}\right)p\left(x_{k}|z_{1:k-1}\right)}{\int_{X}l\left(x_{k}|z_{k}\right)p\left(x_{k}|z_{1:k-1}\right)dx_{k}}. \end{array}$$

The corrected, or posterior, PDF describes the state at step *k* given all available information from observations and predictions at each step.

### 2.2. Particle Filter

The particle filter is a technique which approximates the true distributions of RBE by randomly sampling them. The PF accommodates fully nonlinear motion and observation models and non-Gaussian state belief by propagating and weighting a large set of samples or particles. The first step of the PF is to generate some (preferably large) particles according to an initial belief distribution $p(x_{0})$. These particles are given normalized weights that are usually initially equal. For the prediction stage, each particle is propagated according to the system’s motion model and the weights remain the same. At correction, it is infeasible to move the particles themselves, so instead the weights are reassigned. New weights are calculated by evaluating the observation likelihood $l(x_{k}|z_{k})$ at the locations of the particles and then normalizing them. The prediction stage can then be repeated with the newly weighted particles.

The classic problem with the PF is that, after some iterations, a disparity in particle weights develops. Eventually, most of the particles will negligibly contribute to the distribution while others dominate. To counter this issue, a resampling stage is implemented in which each particle is replaced with a number of evenly weighted particles that is proportional to the original particle’s weight. Resampling at regular intervals usually solves the degeneracy problem and makes the PF a highly effective nonlinear/non-Gaussian estimator.

### 2.3. Zero-Velocity Update Method

IMUs are some of the most commonly used sensors for outdoor human localization because most are small enough to wear. To accomplish human localization by dead-reckoning, IMUs are typically placed on the lower limbs of an individual. The accelerometers in an IMU alone are accurate in determining position changes over small distances, but numerical integration of even the smallest levels of noise creates small biases that accumulate over many measurements. To overcome this drawback, IMU algorithms often implement strap-down integration, which leverages outside information to mitigate drift. To integrate acceleration data twice to obtain the position of a walking individual, the zero-velocity update (ZUPT) method [33] is commonly used. The ZUPT method uses the knowledge that the foot must have zero velocity at regular stance phases in the gait cycle [34,35]. Therefore, the integration error between acceleration and velocity can be eliminated at each stance phase by subtracting off any non-zero velocity that has accumulated since the prior step. This means that the error in the translational velocity of the foot never exceeds that of a single step. However, error will still accumulate during the integration from velocity to position. The only way in which this error can be reduced is with information about global position from another source. Therefore, whenever possible, it is best to combine IMU data with data from a GPS or other external global sensor.

### 2.4. Sensor Fusion

While IMUs can provide locally accurate human position estimation when paired with the ZUPT technique, GPS sensors are generally more accurate over larger distances. This accuracy difference is illustrated in Figure 1, where an IMU is initially more precise than a GPS but eventually IMU uncertainty diverges cubically whereas GPS uncertainty is stationary.

Fusing the sensor data from both these types of sensors yields a much higher accuracy, as shown in [33]. Let ${\bar{x}}_{k}^{I}$ be the estimate of an individual’s 2D (i.e., lat./long.) position at step *k*, as estimated by an IMU *I*, and let $\Sigma_{x_{k}}^{I}$ be the covariance in that estimate. (This can be the output of a real-time ZUPT algorithm.) Similarly, let ${\bar{x}}_{k}^{G}$ be the estimate coming from a GPS *G* at the same (or similar) time step and let $\Sigma_{x_{k}}^{I}$ be the corresponding covariance. It is worth mentioning that the covariance in the IMU estimate will grow with time due to the uncertainty associated with IMU drift, while the GPS covariance is usually nearly constant over a small range of travel. Under the assumption that the IMU and GPS readings are drawn from Gaussian distributions, they can be fused online according to
(3a)$$\begin{array}{c} \Sigma_{x_{k}}^{GI}={\left({\left(\Sigma_{x_{k}}^{I}\right)}^{-1}+{\left(\Sigma_{x_{k}}^{G}\right)}^{-1}\right)}^{-1} \end{array}$$
(3b)$$\begin{array}{c} {\bar{x}}_{k}^{GI}=\Sigma_{x_{k}}^{GI}\left({\left(\Sigma_{x_{k}}^{I}\right)}^{-1}{\bar{x}}_{k}^{I}+{\left(\Sigma_{x_{k}}^{G}\right)}^{-1}{\bar{x}}_{k}^{G}\right) \end{array}$$
where $\Sigma_{x_{k}}^{GI}$ is the covariance of the fused GPS/IMU measurement and ${\bar{x}}_{k}^{GI}$ is the mean of the fused estimate. It was shown in [33] that, if the sensors’ statistics are accurately characterized, the fusion of their measurements will provide an improved estimate of a human’s location.

## 3. Calculation

This section describes the walking trajectory estimation and how the probabilistic step model is implemented.

Figure 2 summarizes this paper’s proposed framework, which we term the mode-finding filter (MFF). The MFF is an RBE estimator and a type of MAP filter, as it maximizes *a posteriori* location belief. It is important to clarify here the distinction between a walking trajectory estimator and an online gait localizer. The former is concerned with reconstructing the path of a walking person, while the latter aims to accurately deliver location information at any time during a walking event. This framework is presented as a real-time localizer, but reconstructing the entire walking trajectory is as simple as just retaining all estimated target locations at each step.

Under the proposed framework, sensor input from a GPS and/or an IMU is fused with a prediction about an individual’s location based on a probabilistic step model. Due to cartesian non-Gaussianity in the step model, this posterior belief function is also non-Gaussian. In order to efficiently reduce this PDF into a point estimate that more accurately summarizes the posterior belief than simply taking its mean, a gradient-ascent step is implemented to find the mode of the distribution. This is then combined with information from the previous step to update the prediction model. These stages are developed in greater detail in the following subsections.

### 3.1. Probabilistic Step Model

The addition of a probabilistic prediction model for walking trajectory estimation where past frameworks have not included such a model can be justified by analogy. Consider a human-supervised localizer, where each measured step vector of an estimation target is confirmed or rejected by an intelligent overseer. The human supervisor may not have any information about where the target is going, but they can still make an informed decision about the accuracy of a step estimate as delivered by one or more intrinsic or extrinsic sensors. This is because the human mind contains a latent model of how humans walk (i.e., each sequential step should be in approximately the same direction as the prior step and the size of each step should be proportional to the length of the average person’s legs). This latent model can be refined with the injection of additional information. For example, if the target of localization is an adult male going on a long hike, the parameters of the latent model will be tuned differently than for a child playing hide-and-seek. This subsection details the probabilistic step model used for the MFF as developed in this paper.

The two principal elements of a step are its size (radius, *r*) and direction (angle, ϕ). Hence, a polar coordinate system is most natural for the probabilistic step model. Consider the following assumptions:An individual’s step sizes while walking a long, quasi-straight path are normally distributed;The direction of those steps is also (locally) Gaussian;The random fluctuations in the direction and size of a step are independent.

If all the above are true, or at least reasonable, then it can be reasonably assumed without much more information that $r_{k}$ and $\phi_{k}$ for the *k*th step may be drawn from a bivariate normal distribution:(4)$$\begin{array}{c} \left[\begin{array}{c} r_{k}\\ \phi_{k} \end{array}\right]=\rho_{k}=N\left(\rho ;{\bar{\rho }}_{k},\Sigma_{\rho_{k}}\right), \end{array}$$
where the normal distribution is given by
(5)$$\begin{array}{c} N\left(x;\bar{x},\Sigma_{x}\right)=\frac{1}{\sqrt{|2\pi \Sigma_{x}|}}\exp\left(-\frac{1}{2}{\left(x-\bar{x}\right)}^{⊤}\Sigma_{x}^{-1}\left(x-\bar{x}\right)\right). \end{array}$$

While this distribution is Gaussian in the step-wise polar coordinate system, it becomes non-Gaussian in the common Cartesian belief space of the localization problem due to the nonlinearity of the conversion from polar to Cartesian coordinates. Figure 3 demonstrates this non-Gaussianity for a single step distribution. While the polar-to-Cartesian conversion could be linearized and thus deliver a purely Gaussian approach, such an approach would only be approximate, and would lose information about the “true” probability density.

The important parameters of the probabilistic step model, then, are ${\bar{\rho }}_{k}$ and $\Sigma_{\rho_{k}}$. Following the third assumption above, we assert that the correlation between *r* and ϕ is negligible (i.e., the covariance matrix $\Sigma_{\rho }$ is diagonal). This leaves only four parameters: the means and standard deviations of *r* and ϕ. The mean angle ${\bar{\phi }}_{k}$ is continuously adapted during the estimation cycle, while the standard deviations $\sigma_{\phi_{k}}$ and $\sigma_{r_{k}}$ are largely context-dependent and learned online. (It is worth mentioning that these parameters could also be adapted in the estimation algorithm, but that is left as future work.)

As mentioned previously, the mean step size is strongly linked to the anatomy of the individual who is being localized. In order to build a robust and realistic model of the expected step size of a walking person, the kinematics of human gait can be exploited.

As Figure 4 demonstrates, a person’s step size depends primarily on the lengths of the upper and lower leg as well as the knee and hip angles during the stance phase when both legs are planted on the ground. Using this planar model of the human body, a single step (defined here as the distance between points where one foot touches the ground) is given by
(6)$$\begin{array}{c} \bar{r}=2l_{l}\left(\sin\left(\theta_{h_{e}}+\theta_{k_{e}}\right)+\sin\left(\theta_{h_{f}}-\theta_{k_{f}}\right)\right)+2l_{u}\left(\sin\left(\theta_{h_{e}}\right)+\sin\left(\theta_{h_{f}}\right)\right). \end{array}$$

Leg lengths can be easily measured, and leg angles have been well-studied in the biomechanics community. Astephen et al. (2008) published statistics on hip and knee angles for 60 healthy subjects obtained by a lab grade motion capture system [36]. They found that the mean range of hip angles over a gait cycle is 39.2${}^{\circ }$± 4.8${}^{\circ }$ and the mean knee angle varies with time, as plotted in Figure 5. Assuming a healthy and symmetrical gait, the left–right knee angles can be matched up by simply shifting the plot by 50% of the gait cycle and drawing a vertical line at any phase of the cycle. The degrees of knee flexion and extension at stance are shown in the figure accordingly. These angles are used in Section 4 both in the realistic generation of simulated steps and in the motion model applied to estimate those steps.

### 3.2. Globalization, Prediction, and Information Fusion

Information from IMU and GPS sensors about walking trajectory is received quite differently. A GPS typically provides individual latitude/longitude readings at relatively slow sample rates (∼1–10 Hz), while an IMU measures accelerations and angular velocities at high frequencies (∼100–1000 Hz), which must be heavily processed to obtain meaningful trajectory information (see Section 2.3). The most natural shared belief space of the GPS and IMU is the 2D (for moderately flat terrain) Cartesian coordinate system whose origin is at the beginning of the estimated path and whose x- and y-axes align with east and north, respectively. In order to incorporate information from the prediction model into the localization framework, it must be transformed into this global coordinate system which is shared with the GPS and/or IMU sensors.

Let the state vector $x_{k}={\left[x_{k}\;y_{k}\right]}^{⊤}$ describe the 2D location of a walking person, where *k* indexes the physical steps that the individual takes. The transition PDF is given, according to the previously established probabilistic step model, in the global coordinate system as
(7)$$\begin{array}{c} p\left(x_{k}|x_{k-1}\right)=p\left(x_{k}-x_{k-1}\right)=N\left(\rho^{\prime }\left(x_{k},x_{k-1}\right);{\bar{\rho }}_{k},\Sigma_{\rho_{k}}\right), \end{array}$$
where
(8)$$\begin{array}{c} \rho^{\prime }\left(x_{k},x_{k-1}\right)=\left[\begin{array}{c} \sqrt{{\left(x_{k}-x_{k-1}\right)}^{2}+{\left(y_{k}-y_{k-1}\right)}^{2}}\\ \tan^{-1}\left(\frac{y_{k}\;-\;y_{k-1}}{x_{k}\;-\;x_{k-1}}\right) \end{array}\right]. \end{array}$$

Recall that this distribution is shown in Figure 3. Once the state transition PDF has been globalized, it must be combined with previous state belief before it can be fused with sensor information.

Recall that the predicted PDF $p(x_{k}|z_{1:k-1})$, defined in Equation (Equation 1), is equivalently given by the convolution integral for Markovian processes with independent increments (as is assumed here). If the initial belief about the target’s location $p(x_{0}|z_{0})$ is a point estimate at ${\hat{x}}_{0}$, it can be modeled by a delta function:(9)$$\begin{array}{c} p\left(x_{0}|z_{0}\right)=\delta \left(x-{\hat{x}}_{0}\right). \end{array}$$

Therefore, the predicted PDF will simply be
(10)$$\begin{array}{c} p\left(x_{1}|z_{0}\right)=N\left(\rho^{\prime }\left(x_{1},{\hat{x}}_{0}\right);{\bar{\rho }}_{0},\Sigma_{\rho_{0}}\right). \end{array}$$

In effect, the first prediction “centers” the distribution of Figure 3 around the initial point estimate of the target location. Furthermore, since the MFF condenses state belief to its mode at each iteration of the filter, reducing the PDF $p(x_{k-1}|z_{1:k-1})$ to just a delta function $\delta (x-{\hat{x}}_{k-1})$, the general expression for the predicted PDF under the MFF is
(11)$$\begin{array}{c} p\left(x_{k}|z_{k-1}\right)=N\left(\rho^{\prime }\left(x_{k},{\hat{x}}_{k-1}\right);{\bar{\rho }}_{k},\Sigma_{\rho_{k}}\right), \end{array}$$
where ${\hat{x}}_{k-1}$ is the mode of the posterior distribution $p(x_{k-1}|z_{1:k-1})$ from the previous step.

Since the proposed framework is a recursive Bayesian estimator, predicted belief must next be fused with sensor observations. Let $l(x_{k}|z_{k}^{G})$ denote an observation likelihood coming from a GPS sensor’s measurement $z_{k}^{G}$ and $l(x_{k}|z_{k}^{I})$ denote an observation likelihood coming from an IMU measurement $z_{k}^{I}$. The MFF algorithm assumes these are both generally Gaussian distributions, though their true probabilistic characterization is context- and sensor-specific. Let the means and covariances of the GPS and IMU likelihoods be ${\bar{x}}_{k}^{G}$, ${\bar{x}}_{k}^{I}$, $\Sigma_{x_{k}}^{G}$, and $\Sigma_{x_{k}}^{I}$, respectively. The fusion of the two Gaussian distributions yields a third Gaussian observation likelihood $l(x_{k}|z_{k}^{GI})$ whose covariance and mean are given by Equation (3). This is an elegant result of the fact that the product of two Gaussians is a third unnormalized Gaussian.

Finally, correction is accomplished by implementing Equation (Equation 2) to obtain $p(x_{k}|z_{1:k})$. It is worth noting that, since the denominator of the correction formula is a constant, it will not affect the location of the mode of $p(x_{k}|z_{1:k})$ and therefore can simply be ignored. The product of the predicted PDF and the GPS/IMU fused observation likelihood is then
(12)$$\begin{array}{c} p\left(x_{k}|z_{1:k}\right)=\left[N(\rho^{\prime }\left(x_{k},{\hat{x}}_{k-1}\right);{\bar{\rho }}_{k},\Sigma_{\rho_{k}})\right]\left[N(x_{k};{\bar{x}}_{k}^{GI},\Sigma_{x_{k}}^{GI})\right]. \end{array}$$

Figure 6 demonstrates the fusion of the two sensors and subsequent correction for a single step. This distribution is generally non-Gaussian in the global Cartesian coordinate system due to the nonlinearities in $\rho^{\prime }$, but it is well-defined.

### 3.3. A Posteriori PDF Maximization via Gradient Ascent

While the corrected, or *a posteriori*, PDF is functionally well-defined, locating its maximum in closed form is intractable. For this reason, a gradient ascent solution is implemented. Equation (Equation 12) can be expanded with the scalar normalization factors of each Gaussian ignored since they do not affect the gradient of $p(x_{k}|z_{1:k})$:(13)$$\begin{array}{c} \begin{array}{c} p\left(x_{k}|z_{1:k}\right)\propto \exp(-\frac{1}{2}[{\left(\rho^{\prime }\left(x_{k},{\hat{x}}_{k-1}\right)-{\bar{\rho }}_{k}\right)}^{⊤}\Sigma_{\rho_{k}}^{-1}\left(\rho^{\prime }\left(x_{k},{\hat{x}}_{k-1}\right)-{\bar{\rho }}_{k}\right)\\ \;\;\;\;\;\;\;\;\;+{\left(x_{k}-{\bar{x}}_{k}^{GI}\right)}^{⊤}{\left(\Sigma_{x_{k}}^{GI}\right)}^{-1}\left(x_{k}-{\bar{x}}_{k}^{GI}\right)]). \end{array} \end{array}$$

The gradient of this distribution is then
(14)$$\begin{array}{c} \nabla_{x_{k}}\left(p(x_{k}|z_{1:k})\right)\propto p\left(x_{k}|z_{1:k}\right)(-\frac{\partial \rho^{\prime }\left(x_{k},{\hat{x}}_{k-1}\right)}{\partial x_{k}}\Sigma_{\rho_{k}}^{-1}\left(\rho^{\prime }\left(x_{k},{\hat{x}}_{k-1}\right)-{\bar{\rho }}_{k}\right)\\ \;\;\;\;\;\;\;\;\;\;\;\;\;-{\left(\Sigma_{x_{k}}^{GI}\right)}^{-1}\left(x_{k}-{\bar{x}}_{k}^{GI}\right)), \end{array}$$
where
(15)$$\begin{array}{c} \frac{\partial \rho^{\prime }\left(x_{k},{\hat{x}}_{k-1}\right)}{\partial x_{k}}=\left[\begin{array}{cc} \frac{x_{k}-{\hat{x}}_{k-1}}{\sqrt{{\left(x_{k}-{\hat{x}}_{k-1}\right)}^{2}+{\left(y_{k}-{\hat{y}}_{k-1}\right)}^{2}}} & \frac{y_{k}-{\hat{y}}_{k-1}}{\sqrt{{\left(x_{k}-{\hat{x}}_{k-1}\right)}^{2}+{\left(y_{k}-{\hat{y}}_{k-1}\right)}^{2}}}\\ -\frac{y_{k}-{\hat{y}}_{k-1}}{{\left(x_{k}-{\hat{x}}_{k-1}\right)}^{2}+{\left(y_{k}-{\hat{y}}_{k-1}\right)}^{2}} & -\frac{y_{k}-{\hat{y}}_{k-1}}{{\left(x_{k}-{\hat{x}}_{k-1}\right)}^{2}+{\left(y_{k}-{\hat{y}}_{k-1}\right)}^{2}} \end{array}\right]. \end{array}$$

The mode of $p(x_{k}|z_{1:k})$ is found by iterating the well-known gradient ascent formula
(16)$$\begin{array}{c} {\hat{x}}_{k}^{i}={\hat{x}}_{k}^{i-1}+\delta^{i-1}\nabla_{x_{k}}\left(p(x_{k}|z_{1:k})\right){|}_{{\hat{x}}_{k}^{i-1}}, \end{array}$$
until there is sufficient convergence in ${\hat{x}}_{k}$. While this will always find the mode of $p(x_{k}|z_{1:k})$ since the PDF is unimodal, the efficiency with which the mode is found strongly depends on the initial estimate ${\hat{x}}_{k}^{0}$. A good candidate for the initial estimate which has been found to work well is proposed as follows.

By linearizing the probabilistic step model, $p(x_{k}|z_{1:k-1})$ can be approximated by a Gaussian distribution which can be fused with the sensor observations in a fashion akin to Equation (3). The mean and covariance of the linearized step model are given by
(17a)$$\begin{array}{c} {\bar{x}}_{k|1:k-1}={\hat{x}}_{k-1}+\bar{r}\left[\begin{array}{c} \cos\left(\bar{\phi }\right)\\ \sin\left(\bar{\phi }\right) \end{array}\right], \end{array}$$
(17b)$$\begin{array}{c} \Sigma_{x_{k|1:k-1}}=\left[\begin{array}{cc} {\left(\bar{r}\sin\left(\bar{\phi }\right)\sigma_{\phi }\right)}^{2}+{\left(\sigma_{r}\cos\left(\bar{\phi }\right)\right)}^{2} & 0\\ 0 & {\left(\bar{r}\cos\left(\bar{\phi }\right)\sigma_{\phi }\right)}^{2}+{\left(\sigma_{r}\sin\left(\bar{\phi }\right)\right)}^{2} \end{array}\right]. \end{array}$$

From this, the recommended initial estimate of ${\hat{x}}_{k}$ is
(18)$$\begin{array}{c} {\hat{x}}_{k}^{0}={\left[\Sigma_{x_{k|1:k-1}}^{-1}+{\left(\Sigma_{x_{k}}^{GI}\right)}^{-1}\right]}^{-1}\left[\Sigma_{x_{k|1:k-1}}^{-1}{\bar{x}}_{k|1:k-1}+{\left(\Sigma_{x_{k}}^{GI}\right)}^{-1}{\bar{x}}_{k}^{GI}\right]. \end{array}$$

This initial estimate is generally close enough to the true mode of the distribution that relatively few gradient ascent iterations are required. Furthermore, using this ${\hat{x}}_{k}^{0}$, the step size δ can usually be assigned a small constant value without concern of inordinately long convergence.

In sum, the MFF for walking trajectory estimation requires only the tracking and handling of means and covariances of sensor measurements and model parameters. Contrary to a Kalman filter, however, the gradient ascent step handles the non-Gaussianity introduced by the probabilistic step model. This gives the filter a balance between accuracy and efficiency that is investigated in the following section.

## 4. Results

The proposed non-Gaussian walking trajectory estimation framework was validated in simulation. This not only provides access to ground-truth for accurate error analysis, but it also allows for the fast generation of thousands of experiments from which important summary statistics can be extracted. The MFF was simulated in parallel with a particle filter since the PF is the industry standard for non-Gaussian estimation, and as such, it is often considered the baseline against which novel filters are compared. The results of the two filters are presented and discussed in the subsections below.

### 4.1. Monte Carlo Simulations

A Monte Carlo approach was taken to assess the performance of the two filters. In each trial, 5000 steps were randomly generated with leg dimensions drawn from the statistics of a set of US Army soldiers. The mean step size was dictated by Equation (Equation 6), while the mean step direction was governed by a low-frequency sinusoidal path with random variation. This choice of path was arbitrary and meant to demonstrate the effectiveness of the MFF on a quasi-straight but still non-trivial walking path. The performance of the framework is independent of the path, so long as consecutive steps are in approximately the same direction.

Table 1 shows the standard values for IMU and GPS bias and covariance used in data collection. These values were obtained from characterization of real sensors. The bias of the IMU represents the average error, in polar coordinates, of the estimate of a step as delivered by an IMU-based ZUPT algorithm. The IMU covariance $\Sigma_{s_{k}}^{I}$ describes the corresponding polar uncertainty of an individual step $s_{k}$. The GPS modeled here was assumed to have no Cartesian bias (i.e., the average of many measurements of a static target equals the target’s true position), but the uncertainty $\Sigma_{x_{k}}^{G}$ was nonzero (i.e., the covariance of measurements of a static target was on the order of 20 meters). It is worth noting that the IMU model was derived from a high-fidelity sensor with relatively high sample rate (APDM, Inc., Portland, Oregon, Opal IMU, 256 Hz), whereas the GPS was modeled after a less expensive sensor with lower rate (Polar^®^ Team Pro, Kempele, Finland, 1 Hz). IMU uncertainty values are much smaller than those of the GPS both because it is a better sensor and because the IMU delivers steps in a local frame rather than locations in a global frame. The covariance $\Sigma_{x_{k}}^{I}$ of the global observation likelihood delivered by the IMU is proportional to $k\Sigma_{s_{k}}^{I}$ because the IMU’s uncertainty increases with time as described in Section 2.3.

The covariances in Table 1 were used as baseline values in a parametric study which assessed the accuracy of the MFF and the PF for various combinations of sensors with different levels of noise. The results of this study are presented qualitatively in Figure 7, Figure 8 and Figure 9 and quantitatively in Table 2. For each combination of $\Sigma_{s_{k}}^{I}$ and $\Sigma_{x_{k}}^{G}$, 5000 steps were simulated and the root mean squared errors (RMSEs) of each signal were computed. Figure 7 shows the results of one such simulation where no filter was used and raw sensor measurements were unaccompanied by the probabilistic step model. As the figure demonstrates, the GPS is an accurate but imprecise sensor, while the IMU trades accuracy for precision. By fusing the two sensors, their strengths are combined to deliver an accurate and precise estimate of the walking trajectory. However, there is still some unavoidable error and lack of smoothness in the fused estimate.

The results of the MFF are shown in Figure 8. As this figure shows, incorporating the step model has a clear smoothing effect on trajectory estimation regardless of which sensors(s) is/are used. Furthermore, the IMU-only drift problem is eliminated. There is, however, oscillatory behavior of the MFF trajectories about ground-truth.

Figure 9 shows the performance of the PF under the same conditions as the MFF. It is worth noting that the PF signals also demonstrate the oscillations about ground-truth since the model is updated in the same way for the PF. However, since the PF retains a fuller estimate of the non-Gaussian state belief distributions at each step, it provides a slightly more accurate trajectory estimate overall.

Table 2 captures the results of these Monte Carlo simulations more quantitatively. The bold values highlight the “winners” for each combination of sensor uncertainties. These results show that incorporating the proposed probabilistic step model significantly improves the accuracy in estimation for both the IMU-only and GPS-only measurements, as reflected in their lower RMSE magnitudes, than the model-less counterparts. However, the model has an unexpectedly negative effect on overall accuracy for the IMU+GPS fused estimate.

Table 2 also communicates the relative performance of the MFF and the PF. The particle filter outperforms the mode-finding filter in all cases except when IMU uncertainty is low and GPS uncertainty is high. Again, this is because the PF retains much more information about the true state belief than the MFF. Nevertheless, the two estimators perform extremely similarly in the majority of cases. This necessitates an investigation of the relative merits of the two filters from the standpoints of both accuracy and efficiency.

### 4.2. Accuracy and Efficiency Trade-Off Study

The first significant difference between the particle filter and the proposed mode-finding filter for walking trajectory estimation is the number of stages required for the two estimators. While the PF requires at least six stages (see Section 2.2), the MFF is much more streamlined since all the stages of RBE are accomplished simultaneously in the gradient ascent algorithm. Figure 10 presents the average computation time for each stage of the two estimators over 5000 steps on a non-dedicated Intel i7 processor in Matlab 2020a. It is worth noting that the initializations of both filters only occur once, while all subsequent stages are executed at every step. According to the figure, the PF is nearly three times as computationally expensive and contains twice the number of stages as the MFF. Of course, these values heavily depend on the number of particles used in the PF and the step size δ chosen for the MFF. For this reason, it is beneficial to examine the dependency of efficiency and accuracy on these two critical parameters.

Using the baseline sensor parameters of Table 1, nine 5000-step simulations were conducted for increasing numbers of particles and decreasing step sizes. The overall RMSEs and total computation times for the two estimators were recorded and plotted in Figure 11 and Figure 12.

As Figure 11 shows, there is little to no correlation between the accuracy of the filters and their “resolution”. This is likely because of the low dimensionality of the belief space (with respect to the PF) and the efficacy of Equation (Equation 18) for initializing gradient ascent (with respect to the MFF). Furthermore, the error of both filters is comparable, though the PF has a slight advantage on average. However, there is a clear exponential relationship between computation time and both the number of particles in the PF and the step size in the MFF. Of chief importance in Figure 12 is the rate of growth. The PF becomes much less efficient at higher resolutions, while the efficiency trade-off under the MFF is not nearly as pronounced.

### 4.3. Summary

Table 3 summarizes the advantages and disadvantages of each sensor combination and fusion method described in this section.

The three main items that deserve further explanation are the oscillatory behavior of the trajectory, the model’s negative effect on accuracy for the IMU+GPS combination, and dimensionality issues with the PF. The oscillatory behavior about the ground truth is likely a product of the simplistic step model-updating stage which only takes the previous step into consideration when updating the expected step direction $\bar{\phi }$. If the next step direction was inputted into the model rather than the previous step direction, it is expected that the oscillatory behavior would be reduced or eliminated. The model’s negative effect on accuracy for the IMU+GPS measurement is assumed to be a combination of the aforementioned simplistic model-updating scheme and an unrealistic overconfidence in the step model. These are elements which could be tuned without too much difficulty to deliver even better walking trajectory estimation (see future work), and while the RMSE of GPS+IMU may be lower, the implications are that the steps are more erratic, which makes it difficult to extract latent information about human motion/biomechanics. Although the resolution trade-off of the PF does not likely have real-time penalties for the 2D localization problem posed here, it is likely that the PF would lose much of its advantage if, for example, the elevation of a walking individual were to be estimated simultaneously with the latitude and longitude. If additional states were to be estimated in this application or this method was applied to other high-dimensional applications, the curse of dimensionality may make the MFF preferable to the PF due to its lower computation.

## 5. Conclusions and Future Work

In conclusion, the proposed mode-finding filter performs nearly as well as a particle filter for solving the non-Gaussian problem of walking trajectory estimation. While the incorporation of a probabilistic step model did not improve estimation performance under either filter when both GPS and IMU sensors were used, there was a marked improvement when sensors were implemented individually. The most significant finding is that, while there is no substitute for a diverse sensor suite, substantial improvement in gait-based human localization (and therefore walking trajectory estimation) can be achieved when only one sensor type is available by simply incorporating the proposed probabilistic step model. This no-cost improvement is afforded by the intuition contained in the model of what a "normal" human step looks like. The proposed non-Gaussian MFF balances accuracy with efficiency, retaining only the most important information about state belief at each step. Several Monte Carlo simulation studies validated the MFF against the industry standard PF and established its legitimacy as a comparable MAP-type estimator of walking trajectories.

It is expected that tuning the step model and the mechanism by which it is updated could increase the accuracy of the framework. For example, a more robust model-updating stage may assign the next expected step direction and its uncertainty based on more than just the single previous step. This is a direction of future work which the authors plan to pursue. Furthermore, an obvious next step would be to validate the framework on real experimental data, first offline and then in a real-time context. Such a real-time walking-based human localizer could have applications in *in natura* biomechanics measurement, hazardous military and civilian field work scenarios, cooperative human–robot missions, and even localization of bipedal humanoid robots.

## Figures and Tables

**Figure 1 sensors-23-06494-f001:**
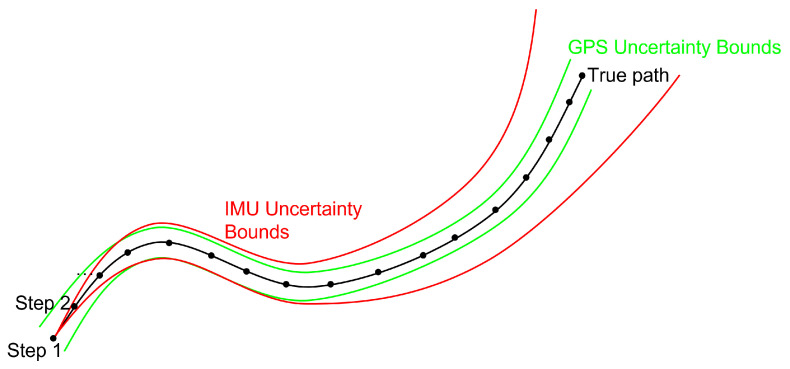
IMU and GPS uncertainty over time.

**Figure 2 sensors-23-06494-f002:**
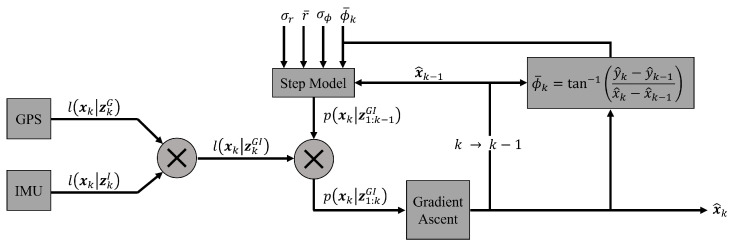
Diagram of proposed framework.

**Figure 3 sensors-23-06494-f003:**
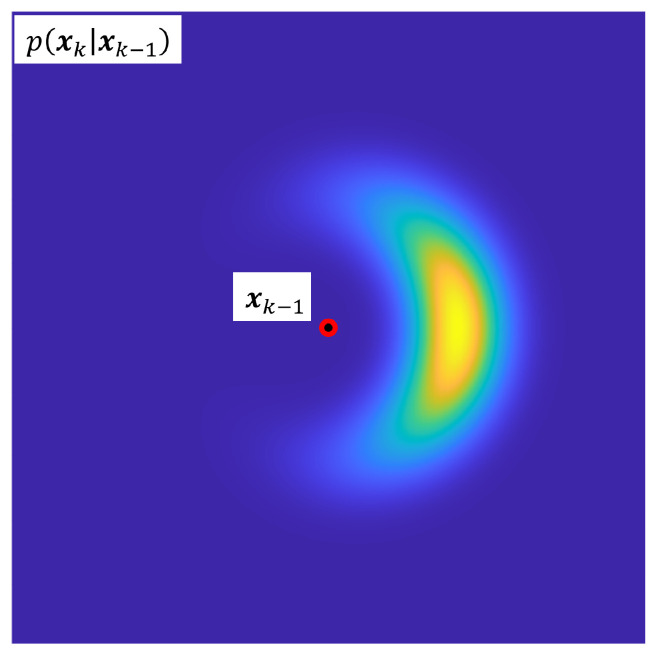
Non-Gaussian probabilistic step model.

**Figure 4 sensors-23-06494-f004:**
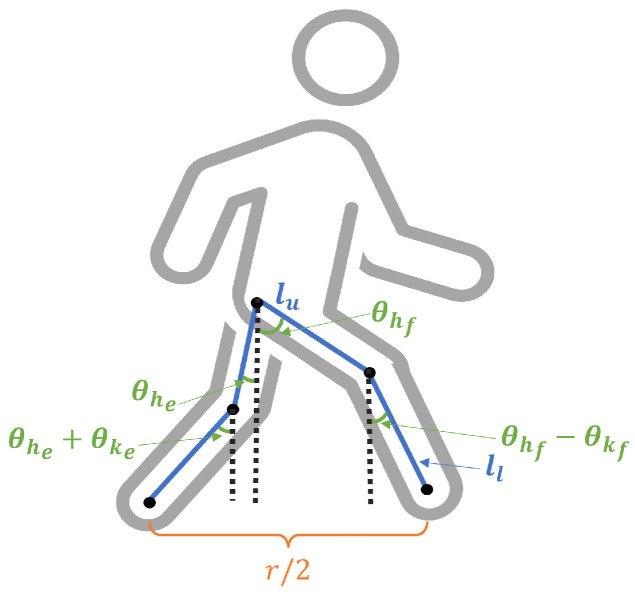
Walking kinematic model. Blue lines represent leg skeletal segments.

**Figure 5 sensors-23-06494-f005:**
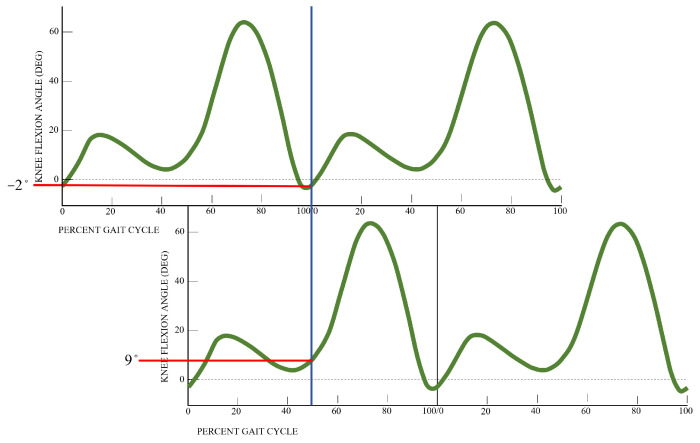
Average knee angles as a percentage of the gait cycle. Blue line represents angles chosen at stance.

**Figure 6 sensors-23-06494-f006:**
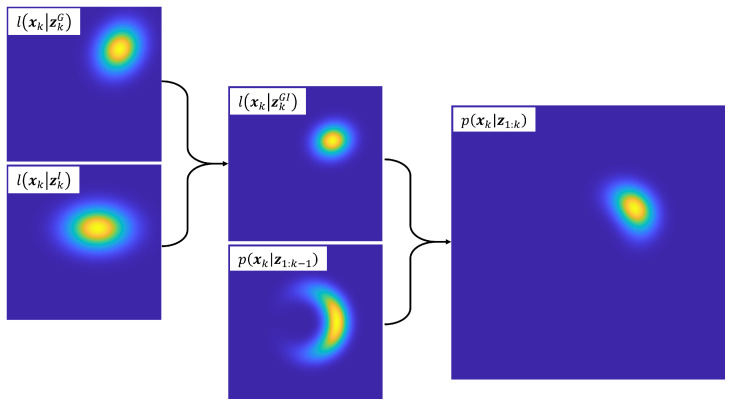
Visualization of observation fusion and correction.

**Figure 7 sensors-23-06494-f007:**
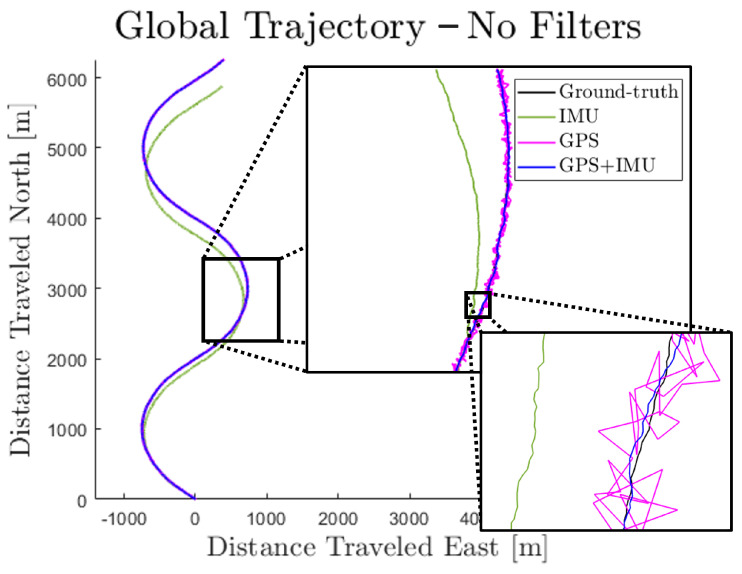
Sample simulation result—no filters.

**Figure 8 sensors-23-06494-f008:**
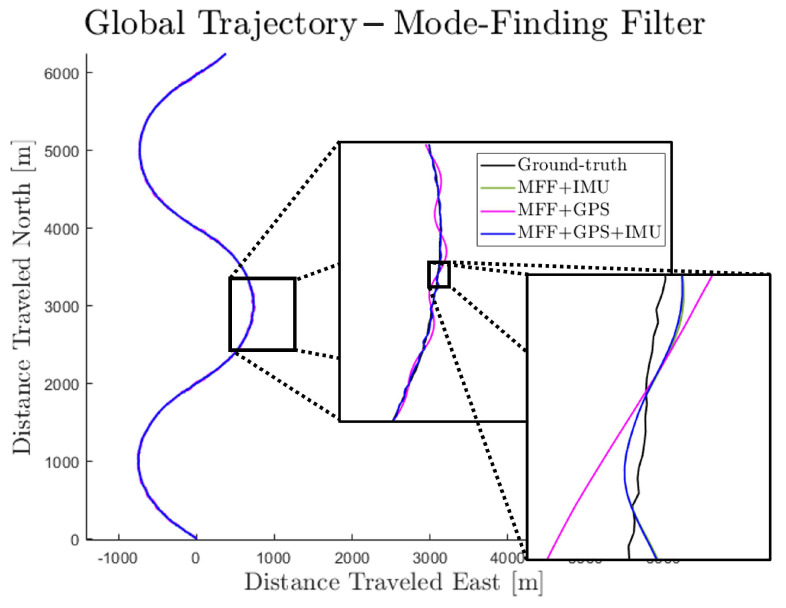
Sample simulation result—mode-finding filter.

**Figure 9 sensors-23-06494-f009:**
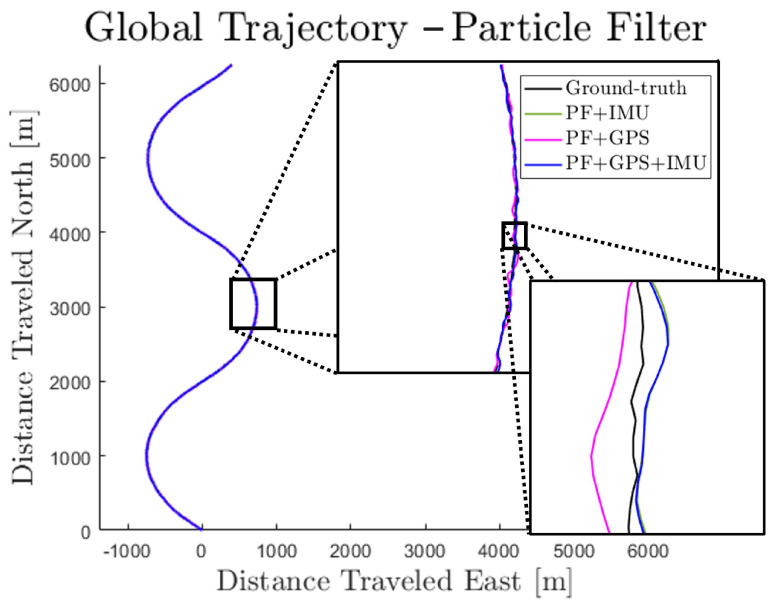
Sample simulation result—particle filter.

**Figure 10 sensors-23-06494-f010:**
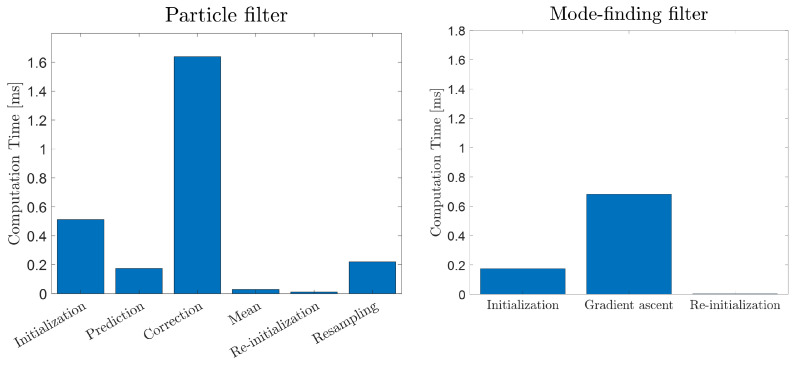
Computation time for the PF (1000 particles) and MFF ($\delta =2^{-4}$).

**Figure 11 sensors-23-06494-f011:**
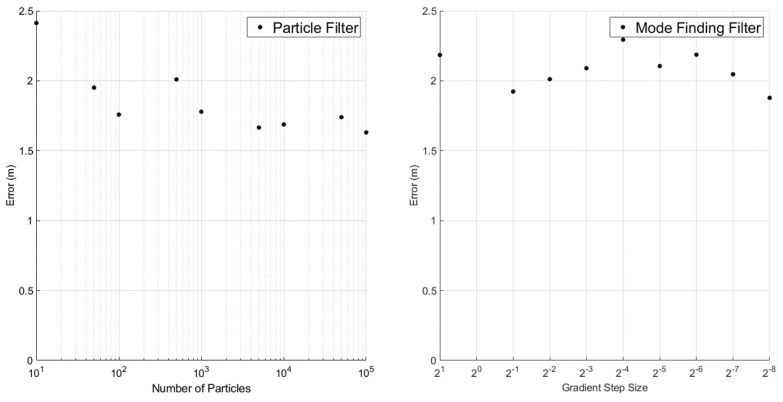
Estimation error vs. number of particles and gradient ascent step size.

**Figure 12 sensors-23-06494-f012:**
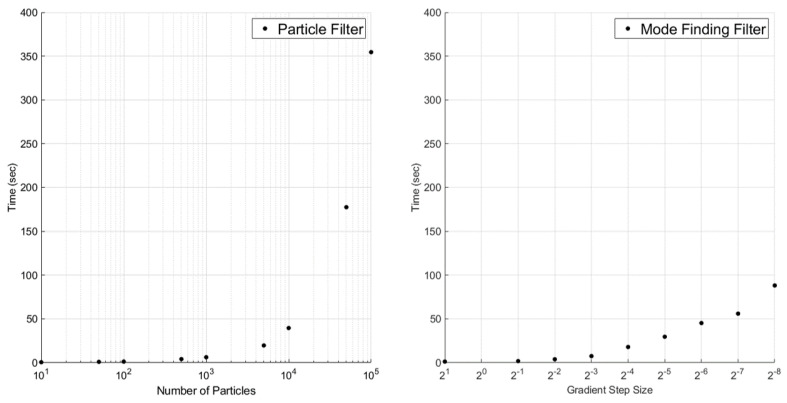
Computation time vs. number of particles and gradient ascent step size.

**Table 1 sensors-23-06494-t001:** Standard bias and covariance values.

	IMU	GPS
Bias	$\left[\begin{array}{c} -0.0531\;m\\ 0.000193\;\mathrm{rad} \end{array}\right]$	$\left[\begin{array}{c} 0\;m\\ 0\;m \end{array}\right]$
Covariance	$\Sigma_{s_{k}}^{I}=\left[\begin{array}{cc} 0.0618 & 0.0018\\ 0.0018 & 0.0532 \end{array}\right]$	$\Sigma_{x_{k}}^{G}=\left[\begin{array}{cc} 21.846 & 0\\ 0 & 24.445 \end{array}\right]$

**Table 2 sensors-23-06494-t002:** RMSE values for each sensor–filter combination (m). Bold values represent most accurate filter and sensor combination for each sensor condition.

**1/4** $\Sigma_{s_{k}}^{I}$	**1/4** $\Sigma_{x_{k}}^{G}$	**1/2** $\Sigma_{x_{k}}^{G}$	$\Sigma_{x_{k}}^{G}$	**2** $\Sigma_{x_{k}}^{G}$	**4** $\Sigma_{x_{k}}^{G}$
IMU	110.426	125.644	111.644	128.324	123.102
IMU+MFF	0.928	1.128	1.278	1.758	1.981
IMU+PF	0.839	1.035	1.202	1.582	1.794
GPS	3.108	4.422	6.268	8.756	142.469
GPS+MFF	2.894	3.554	5.941	8.933	11.332
GPS+PF	1.719	2.439	3.950	6.339	12.759
GPS+IMU	**0.776**	**0.988**	**1.169**	**1.513**	**1.734**
GPS+IMU+MFF	0.905	1.109	1.260	1.745	1.965
GPS+IMU+PF	0.817	1.015	1.190	1.564	1.780
1/2$\Sigma_{s_{k}}^{I}$	1/4$\Sigma_{x_{k}}^{G}$	1/2$\Sigma_{x_{k}}^{G}$	$\Sigma_{x_{k}}^{G}$	2$\Sigma_{x_{k}}^{G}$	4$\Sigma_{x_{k}}^{G}$
IMU	164.980	140.749	144.114	143.172	137.359
IMU+MFF	1.322	1.398	1.599	2.014	2.950
IMU+PF	1.197	1.236	1.403	1.702	2.543
GPS	3.099	4.377	6.199	8.935	12.328
GPS+MFF	2.716	3.979	5.634	8.569	11.660
GPS+PF	1.803	2.568	3.741	5.398	14.148
GPS+IMU	**1.088**	**1.159**	**1.312**	**1.619**	**2.392**
GPS+IMU+MFF	1.239	1.364	1.549	1.971	2.907
GPS+IMU+PF	1.134	1.202	1.372	1.674	2.511
$\Sigma_{s_{k}}^{I}$	1/4$\Sigma_{x_{k}}^{G}$	1/2$\Sigma_{x_{k}}^{G}$	$\Sigma_{x_{k}}^{G}$	2$\Sigma_{x_{k}}^{G}$	4$\Sigma_{x_{k}}^{G}$
IMU	186.437	173.919	165.588	185.337	149.041
IMU+MFF	1.679	1.764	2.202	2.598	2.965
IMU+PF	1.430	1.496	1.835	2.157	2.285
GPS	3.133	4.396	6.213	8.793	12.436
GPS+MFF	3.030	4.191	5.945	8.339	9.092
GPS+PF	1.852	2.469	3.650	6.045	9.035
GPS+IMU	**1.273**	**1.379**	**1.683**	**1.977**	**2.119**
GPS+IMU+MFF	1.531	1.669	2.135	2.546	2.916
GPS+IMU+PF	1.329	1.430	1.779	2.100	2.249
2$\Sigma_{s_{k}}^{I}$	1/4$\Sigma_{x_{k}}^{G}$	1/2$\Sigma_{x_{k}}^{G}$	$\Sigma_{x_{k}}^{G}$	2$\Sigma_{x_{k}}^{G}$	4$\Sigma_{x_{k}}^{G}$
IMU	256.671	268.211	262.768	251.444	238.352
IMU+MFF	2.153	2.583	3.452	4.150	4.539
IMU+PF	1.725	1.982	2.644	3.146	3.560
GPS	3.111	4.383	6.312	8.746	12.296
GPS+MFF	2.777	4.346	5.866	7.419	10.029
GPS+PF	1.790	2.512	4.015	6.799	9.307
GPS+IMU	**1.455**	**1.718**	**2.286**	**2.760**	**3.221**
GPS+IMU+MFF	1.885	2.360	3.217	3.918	4.415
GPS+IMU+PF	1.552	1.844	2.501	3.029	3.480
4$\Sigma_{s_{k}}^{I}$	1/4$\Sigma_{x_{k}}^{G}$	1/2$\Sigma_{x_{k}}^{G}$	$\Sigma_{x_{k}}^{G}$	2$\Sigma_{x_{k}}^{G}$	4$\Sigma_{x_{k}}^{G}$
IMU	352.261	436.073	316.520	444.468	365.252
IMU+MFF	2.404	3.561	3.619	5.288	6.235
IMU+PF	1.858	2.778	2.688	4.203	4.692
GPS	3.104	4.354	6.186	8.911	12.406
GPS+MFF	2.711	4.110	5.792	8.191	10.713
GPS+PF	1.621	2.503	3.347	7.061	10.517
GPS+IMU	**1.520**	**2.136**	**2.209**	**3.340**	**3.870**
GPS+IMU+MFF	2.032	3.072	3.370	4.877	5.952
GPS+IMU+PF	1.632	2.443	2.516	3.890	4.491

**Table 3 sensors-23-06494-t003:** Summary of results.

Measurement	Computation	Accuracy
IMU	Low	Very Inaccurate
IMU+MFF	Low	Somewhat Accurate
IMU+PF	Low	Somewhat Accurate
GPS	Very low	Somewhat Inaccurate
GPS+MFF	Very low	Somewhat Inaccurate
GPS+PF	Very low	Somewhat Inaccurate
GPS+IMU	Low	Most Accurate
GPS+IMU+MFF	Moderate	More Accurate
GPS+IMU+PF	Highest	More Accurate
**Measurement**	**Pros**	**Cons**
IMU	Easy to implement	Very inaccurate
IMU+MFF	Good balance between ease of implementation and accuracy	Neglects GPS data
IMU+PF	Good balance between ease of implementation and accuracy	Neglects GPS data
GPS	Easy to implement	Inacurate, neglects IMU data
GPS+MFF	More accurate than GPS-only	Neglects IMU data
GPS+PF	More accurate than GPS-only	Neglects IMU data
GPS+IMU	Accuracy	Trajectory is not necessarily intuitive
GPS+IMU+MFF	Trajectory has smoothing effect, less computation than PF	Slightly less accurate than PF, RMSE is reduced when model is implemented
GPS+IMU+PF	Trajectory has smoothing effect, lower RMSE than MFF	Higher computation than MFF

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
