# Peer review of "Walking Trajectory Estimation Using Multi-Sensor Fusion and a Probabilistic Step Model"

_sensors, 2023, doi:10.3390/s23146494_

Round 1

Reviewer 1 Report

The submitted manuscript is entitled "Walking Trajectory Estimation Using Multi-Sensor Fusion and a Probabilistic Step Model.

The manuscript describes a very interesting outdoor gait analysis method using GPS and IMU together, and I predict that this research will make a significant contribution to new gait analysis tools.

After reviewing the submitted manuscript, the authors should consider revising the manuscript to address the following points.

#1 Product information for IMU sensor devices and GPS sensor devices

Please provide information on the sensors used for the benefit of researchers conducting subsequent studies to this study. Product name, manufacturer, measurement frequency, etc.

#2 Software used for analysis

As with #1, please provide information on the software used in the analysis for the benefit of researchers who will conduct subsequent studies in this research.

Reviewer 2 Report

Comments on the paper: “Walking trajectory estimation using multi-sensor fusion and a probabilistic step model”

Comment 1: It would greatly improve the clarity of this paper if the authors could accentuate the contributions of their work within the abstract. The current abstract mentions that the proposed method performs similarly to a particle filter in terms of accuracy, and outperforms it in terms of efficiency at higher resolutions. However, the abstract could benefit from a more comprehensive and explicit summary of the study. Specifically, a well-written abstract should answer the following questions: (i) What was the research objective or the issue under investigation?,  (ii) What did the study entail, and what methods or procedures were employed? (iii) What were the primary results or findings of the study? And (iv) What conclusions or implications can be drawn from the work?. Given that abstracts typically range between 250 to 300 words, there appears to be ample room to expand and address these questions more fully. 

Comment 2: In lines 19-21, the authors suggest that no existing standards covers robotic safety in workplace settings. However, it's important to highlight that ISO 15066 specifically delineates safety requirements for human-robot collaboration within industrial environments, with a particular focus on collision avoidance between humans and robots. Yet, this standard's applicability is limited to robotic arms and does not extend to mobile robots, I recommend that the authors adjust their statements to accurately reflect the current state of safety standards in the field.

Comment 3: The authors are advised to elucidate the limitations associated with the Gaussian approach for walking trajectory estimation. Concurrently, they should substantiate why the non-Gaussian perspective is deemed a more appropriate choice in this context. It would greatly enhance the paper's credibility if they could support this viewpoint with relevant bibliographic references.

Comment 4: The authors would benefit from conducting an extensive review of the current state of the art. This should particularly focus on studies that propose fusion algorithms for GPS and IMU, both within Gaussian and non-Gaussian contexts, as well as those considering Bayesian and non-Bayesian methodologies. While it's not within my purview as a reviewer to recommend specific studies, I would encourage the authors to undertake a comprehensive literature search on SCOPUS using the search phrase “Bayesian AND IMU AND GPS” to ensure a thorough understanding and discussion of existing work in this area.

Comment 5: The simulated walking trajectory presented in the study, spanning 6 km in length and 2 km in width, appears disproportionately large for a typical industrial environment involving human-robot collaboration. To enhance the credibility of the results sections, I suggest that the authors consider utilizing GPS and IMU data from a smartphone, carried by an individual traversing a path marked by visible indicators such as yellow ground markers. This approach would likely result in a more realistic simulation more akin to actual industrial conditions

Given the concerns identified during this review process, which include an incomplete abstract, unclear presentation of the paper's main contribution, an insufficiently thorough exploration of the existing literature, and an inadequately presentation of results, I am recommending that this paper be rejected in its current form. However, I would like to encourage the authors to take these observations into account and work towards improving these aspects. I believe that by addressing these areas of concern, the paper could significantly improve in clarity, depth, and overall impact. Once these issues have been addressed, I recommend that the authors consider resubmitting the paper.

Reviewer 3 Report

This paper presents a framework for accurately and efficiently estimating a humans walking trajectory using inertial and/or satellite sensors by incorporating a probabilistic step model. The introduction of the step model creates a non-Gaussian estimation problem which is solved efficiently using a maximum-a-posteriori-type filter. However, several shortcomings are listed as follows.

1) The figures are not in order. Please replace the figures.

2) Formula 6 may not accurately describe human posture, because the hip and knee angles in the walking kinematic model are hard to represent a humans walk.

3) The banana shape of PDF in Fig. 3 is a classical. Please check the reference "Probabilistic models of dead-reckoning error in nonholonomic mobile robots," IEEE International Conference on Robotics and Automation 2003, pp. 1594-1599. G.S. Chirikjian, A.B. Kyatkin, Engineering Applications of Noncommutative Harmonic Analysis, CRC Press, Boca Raton, 2001. If the figure is not your primitive creation, the citations should be added. The latest sensor fusion method is also suggested to add. "Robotic Object Perception Based on Multispectral Few-Shot Coupled Learning," in IEEE Transactions on Systems, Man, and Cybernetics: Systems, doi: 10.1109/TSMC.2023.3279023. "Human-Exploratory-Procedure-Based Hybrid Measurement Fusion for Material Recognition," in IEEE/ASME Transactions on Mechatronics, vol. 27, no. 2, pp. 1093-1104, April 2022, doi: 10.1109/TMECH.2021.3080378.

The Quality of English Language is OK.

Round 2

Reviewer 2 Report

Dear authors, next time please write a response letter addressing point by point the observations made to the article.